

# Molecular evidence for cross boundary spread of *Salmonella* spp. in meat sold at retail markets in the middle Mekong basin area

Dethaloun Meunsene[1], Thanaporn Eiamsam-ang[1], Prapas Patchanee[2], Ben Pascoe[3], Phacharaporn Tadee[4] and Pakpoom Tadee[2]

[1] Graduate Program in Veterinary Science, Faculty of Veterinary Medicine, Chiang Mai University, Muang, Chiang Mai, Thailand
[2] Integrative Research Center for Veterinary Preventive Medicine, Faculty of Veterinary Medicine, Chiang Mai University, Muang, Chiang Mai, Thailand
[3] The Milner Centre for Evolution, University of Bath, Claverton Down, Bath, United Kingdom
[4] Faculty of Animal Science and Technology, Maejo University, San Sai, Chiang Mai, Thailand

Corresponding author
Pakpoom Tadee,
pakpoom.t@cmu.ac.th,
d.pakpoom@gmail.com

## ABSTRACT

**Background**. The surrounding areas of the middle Mekong basin, particularly along the border between Thailand and Lao People's Democratic Republic (Lao PDR), are high-risk areas for many livestock-associated foodborne illnesses, especially salmonellosis. This study aimed to determine the prevalence and characteristics of *Salmonella* spp. contamination in pork, beef and chicken meats sold at retail markets in the Thailand-Laos border area surrounding the Thai-Lao Friendship Bridge I from January to May 2019. We focused on the prevalent serotypes, antimicrobial susceptibility profiles and the multilocus sequence type (MLST) genotypes of the collected *Salmonella* strains.
**Results**. From a total of 370 meat samples collected, 63% were positive for *Salmonella*, with the prevalence of 73%, 60% and 56% from pork, beef and chicken meat samples, respectively. Of all the positive samples, 53 serotypes were identified. Of these, *Salmonella enterica* serovar London accounted for the majority (27%), followed by serovars Corvallis (14%), and Rissen (6%). Resistance against tetracycline was found at the highest frequency (50%), followed by ampicillin (35%) and sulfamethoxazole-trimethoprim (28%). MLST revealed no evidence of shared genetic relatedness of *Salmonella* at retail sites among Thailand-Laos border zone. However, a diverse range of *Salmonella* genotypes were spread over the area. Besides, the persistence of the residential pathogen and sharing of the supply route within-country can be inferred.
**Conclusions**. Given the high levels of contamination of retail meats, regular disinfecting of all working areas and quality control checking at pre-retail stage must be applied to reduce the transmission of *Salmonella* and other foodborne pathogens to consumers. The findings of this study will make a significant contribution to the current understanding of *Salmonella* epidemiology to enhance food security in the region.

## INTRODUCTION

*Salmonella* spp. is a significant causative agent of bacterial foodborne illnesses in humans and can be found worldwide (*Sirichote et al., 2010*; *Van Boxstael et al., 2012*; *Basler et al., 2016*). Annually, ninety million cases resulting in 150,000 deaths among salmonellosis patients have been recorded (*Campioni, Bergamini & Falcao, 2012*). Along with the direct effects *Salmonella* spp. have on the gastrointestinal tract, evidence of drug resistance is also a major public health concern (*Foley & Lynne, 2008*; *Kurtz, Goggins & McLachlan, 2017*; *Jajere, 2019*). This can result in a reduction in the effectiveness of first line empirical treatments and limit treatment choices (*Van Boxstael et al., 2012*).

Livestock products (farm animal-origin food), especially meats, are an important source of human salmonellosis (*Heyndrickx et al., 2002*; *Mainali et al., 2009*; *Rostagno & Callaway, 2012*). Retail markets have been identified as the most significant point of contact for salmonellosis exposure and transmission among humans (*Hauser et al., 2011*; *Gomes-Neves et al., 2012*). Improper management and biosecurity during the production process, such as on farms or in slaughterhouses, also contribute to the risk of increased pathogen loads in retail meats. Contamination can occur directly or through contaminated equipment due to improper handling practices or unsuitable storage conditions (*Lo Fo Wong et al., 2002*).

Increased demand for meat consumption has led to an intensive transformation of the animal production industry (*Guardabassi, Jensen & Kruse, 2008*). Good Management Practices (GMP) must be implemented at all production levels to ensure food safety for consumers. However, these practices are difficult to implement in developing regions, such as the middle Mekong basin and the surrounding areas, particularly along the border between Thailand and Lao People's Democratic Republic (Lao PDR). Animal farming practices are predominantly traditional free-range and smallholder backyard systems, which often employ the minimum hygienic sanitation systems. Moreover, inadequate practices in slaughterhouses and retail outlets, such as on-floor slaughtering and uncontrolled storage conditions in purchasing areas are not ideal practices. Cultural preferences and a lack of awareness among local people about raw meat consumption, in addition to an absence of reliable and high-quality resources, such as clean water and cooking supplies are also important risk factors to be considered (*Wilson, 2007*; *Conlan et al., 2014*; *Okello et al., 2017*). Taken together, these risk factors increase the opportunity for several foodborne diseases, including salmonellosis (*Mughini-Gras et al., 2014*; *Ferrari et al., 2019*).

Increased population density strongly correlates with the risk of infectious disease, exponentially (*Wilkinson et al., 2018*). Large scale population movement in specific geographical areas provide a pathway for disseminating and continuing to shape the epidemic of cross boundary disease (*Muñoz Ramirez et al., 2021*). Volume, speed and reach of the movements have been considered as the potent force contribution (*Wilson, 1995*). In 1994, the Thai-Lao Friendship Bridge I linking Nong Khai Province, Thailand and the Vientiane Capital of Lao PDR was officially opened. More than three million people cross the bridge each year (*Australian Embassy Thailand, 2011*). These people cross by way of private and public vehicles to engage in trade and exchange various products. Therefore,

the chances of pathogen transmissions between borderline communities are likely to be high.

In a study of *Salmonella* prevalence in the meat being sold around the Thailand-Laos border region, *Sinwat et al. (2016)* reported that the prevalence of pork contamination in border areas of Thailand and Laos PDR was 65%. Moreover, *Boonmar et al. (2013)* reported that the prevalence of contaminated pork and beef in southern-Laos PDR was 93% and 82%, respectively. Both studies reported a widespread variety of serotype distribution including *S*. Typhimurium, *S*. Derby, *S*. Anatum and *S*. Rissen. Interestingly, almost all strains of *Salmonella* are currently classified as multidrug-resistant strains. However, based on the data that has been studied, there is an evident lack of reporting information on poultry meat, while it is well known that poultry meat is commonly consumed in this area. Additionally, to gain access to in-depth epidemiological information, a study of the pathogen characteristics at the genetic level should be fulfilled to expand upon the scope of epidemiological knowledge of these pathogens. Multilocus sequence typing (MLST) technique is one of the methods employed in our study. This technique relies on a comparison of the sequences of allele types in a specific group for each set of house-keeping gene and focuses on the genetics to assemble data of the sequence type (ST). The resulting findings can be compared with the information of various databases related to the study of the global bacterial dynamic distribution and their genetic evolution (*Liu et al., 2011*; *Patchanee et al., 2017*; *Zhou et al., 2020*).

The purpose of the study was to investigate the prevalence of *Salmonella* in meat sold at retail markets in the middle Mekong Basin area along the border of Thailand and Lao PDR, surrounding the Thai-Lao Friendship Bridge I. We compared strain serotypes, genotypes and their antimicrobial-resistance patterns and identified genetic and phenotypic variation in *Salmonella* strains. Understandings of the local population structure, their characteristics and the transmission dynamics of the pathogen will inform regional knowledge gaps in the epidemiology and form the basis for appropriate treatment options as well as preventive measures to help control the spread of human salmonellosis in the region.

## MATERIALS AND METHODS

### Overview of experimental program

A schematic diagram of workflow with laboratory procedure used in this study was displayed in Fig. 1.

Source and sample size determination: Marketplaces were purposively selected. All meat for consumption including pork, chicken, and beef samples were randomly collected in aseptic manipulation.

Sample collection: All meat samples were sampled and labeled in the purchasing process. Collector immediately placed each sample in cooler box and transported it to the laboratory within 24 h after collection.

Laboratory procedure: *Salmonella* spp. was detected based on the ISO-6579 standard method. All *Salmonella* positive samples were then serotyped and antimicrobial susceptible tested. Selected *Salmonella* strains were further analyzed by MLST, and phylogenetic tree analysis was finally constructed.

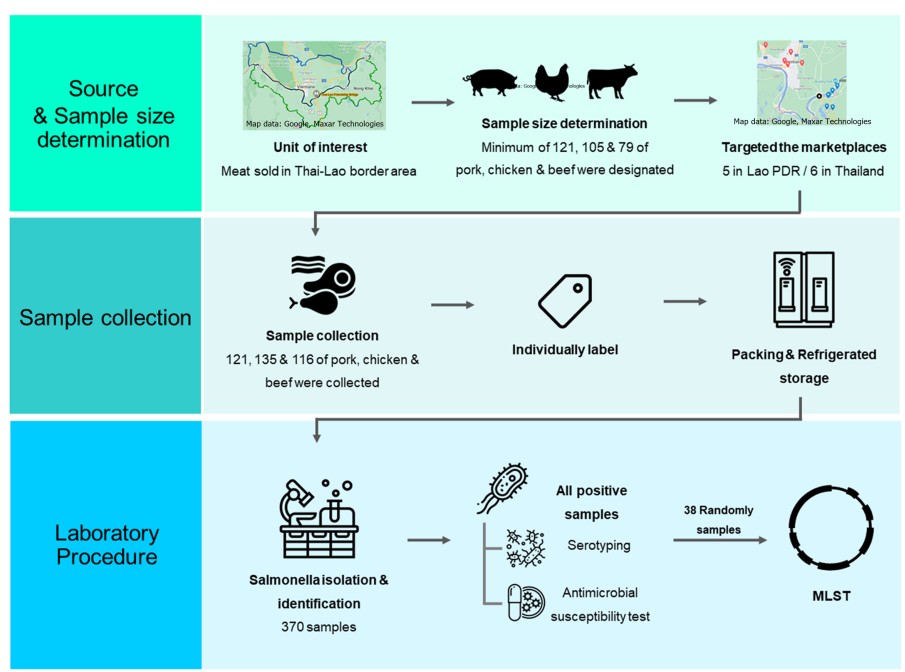

**Figure 1** **A schematic flow diagram of the entire study.**

## Sample collection procedures

The sample size of this study was determined using the Win Epi online program (http://www.winepi.net/uk/index.htm). The prevalence rates identified in previous studies of 65% (*Sinwat et al., 2016*), 73% (*Trongjit et al., 2017*) and 82% (*Boonmar et al., 2013*) were chosen as the "expected prevalence" to calculate the sample sizes for pork, chicken and beef, respectively. An accepted error rate of 8.5% and 95% confidence levels were selected for the required feature inputs. For an infinite population, a minimum of 121, 105 and 79 samples of pork, chicken and beef were designated, respectively. However, in order to achieve greater levels of accuracy and reliability, additional samples were carefully chosen.

During the period of January to May 2019, 370 samples (121, 133 and 116 samples of pork, chicken and beef, respectively), approximated of 200 g/each, were purchased from five retail markets in Vientiane Capital, Lao PDR and six retail markets in Nong Khai Province, Thailand. All targeted markets were selected by convenience sampling. Map of the locations generated by https://www.google.com/maps was displayed in Fig. 2. All samples were individually labeled, put into plastic packs and stored in an icebox for laboratory analysis within 24 hr at the Department of Veterinary Medicine, Faculty of Agriculture, National University of Laos, Nabong Campus, Lao PDR.

## *Salmonella* isolation and identification

Isolation and identification of *Salmonella* spp. from meat samples (pork, chicken, beef) were performed following ISO6579: 2002 Amendment 1:2007, Annex D technique. Accordingly, 25 g of samples were enriched with 225 ml of Buffered Peptone Water (BPW; Merck,

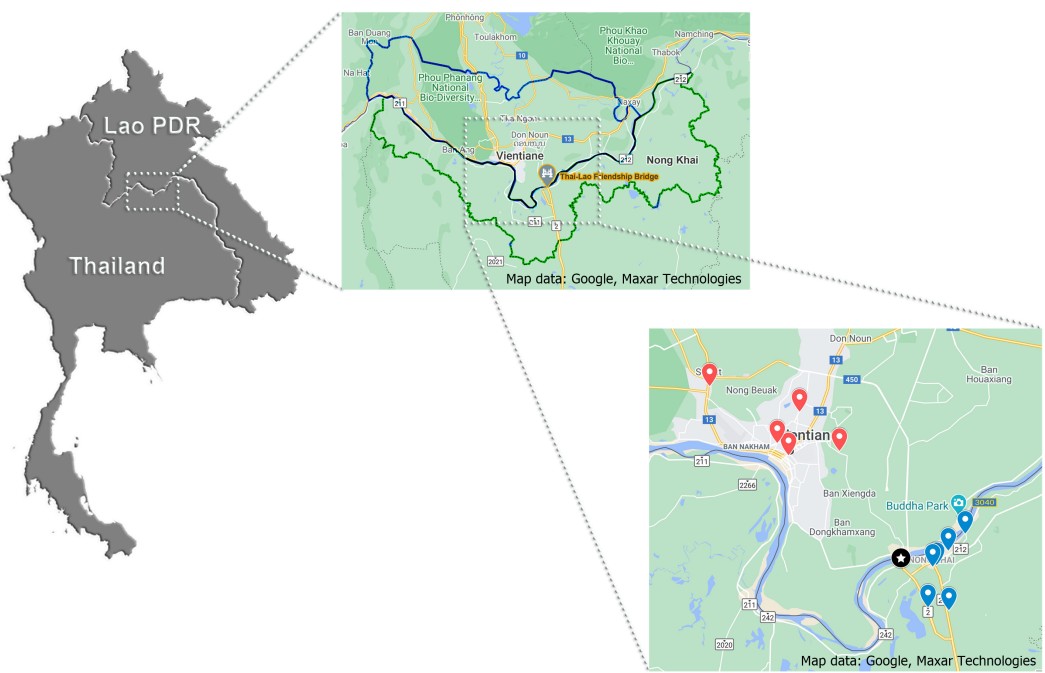

**Figure 2 Geographic location of targeted retail markets of this study.** For the largest scale: Star marking displays the location of Thai-Lao Friendship Bridge I; Blue and red dropped pins distribute the location of targeted Lao and Thai markets, respectively.

Germany). The mixing materials were homogenized for 120 s and incubated at 37 ° C for 24 hr. An aliquot of 100 µl of the pre-enriched inoculum was transferred to 10 ml of Rappaport-Vassiliadis broth (RV; Oxoid, UK) at 42 °C for an incubation period of 24 hr. The likely colonies were then streaked on Xylose-Lysine-Desocholate agar (XLD; Oxoid, UK) and Brilliant-green Phenol Red Lactose sucrose agar (BPLS; Merck, Germany). After an incubation period of 24 hr at 37 °C, the presumptive colonies were then placed on Triple Sugar Iron agar (TSI; Oxoid, England), urease and Motility Indole-Lysine agar (MIL; Merck, Germany) for bio-chemical confirmation. An analysis of the strains indicated a correct bio-chemical reaction as positive for *Salmonella*. Finally, all results were then recorded.

## Serotyping and Antimicrobial susceptibility testing

All detected *Salmonella* spp. specimens were serotyped using the serum-agglutination test according to the White-Kauffmann-Le Minor scheme (*Popoff, Bockemühl & McWhorter-Murlin, 1993*), and were also run through antimicrobial susceptibility testing with agar disk diffusion using ten panels of antimicrobial agents. Amoxicillin-clavulanic acid (AMC) 20/10 µg, ampicillin (AMP) 10 µg, chloramphenicol (C) 30 µg, ciprofloxacin (CIP) 5 µg, cefotaxime (CTX) 30 µg, nalidixic acid (NA) 30 µg, norfloxacin (NOR) 10 µg, streptomycin (S) 10 µg, sulfamethoxazole-trimethoprim (SXT) 23.75/1.25 µg and tetracycline (TE) 30 µg (*CLSI, 2011*) were included. Strains that existed with intermediate resistance were

grouped as being susceptible in order to avoid overestimation. Strains that resisted ≥ 3 of the antimicrobial agents were considered multidrug-resistant.

## Multilocus sequence typing (MLST)

Randomly selected 38 specimens of *Salmonella* obtained from the most frequently found serotypes were genotyped using the MLST technique. DNA was extracted according to the protocol described by *Liu et al. (2011)*. Seven housekeeping genes, including *aro*C (chorismate synthase); *dna*N (DNA polymerase III beta subunit); *hem*D (uroporphyrinogenIII cosynthase); *pur*E (phosphoribosylaminoimidazole carboxylase); *suc*A (alpha ketoglutarate dehydrogenase); *his* D (histidinol dehydrogenase) and *thr*A (aspartokinase I/homoserine dehydrogenase), were selected for MLST profiling (REF). PCR amplification of all 7 genes was accomplished using the method previously described by *Kotetishvili et al. (2002)*. In brief, the PCR amplification conditions were 94 ° C for 5 min, followed by 35 amplification cycles, each consisting of sequential incubation at 94 °C (45 s), 55 °C (45 s), and 72 °C (5 min). Subsequently, the products were sent to be sequenced at the Macrogen Service Center, Republic of Korea.

The sequences obtained in each gene (File S1 and Table S1) were transformed into allele numbers, and were compiled according to the sequence type (ST) data obtained from the http://enterobase.warwick.ac.uk/species/senterica/allele_st_search database. Finally, all data of the *Salmonella* strains acquired from this study were analysed. A phylogenetic tree was constructed using Bionumerics® software version 7.6 (Applied Maths, Belgium).

## Statistical analysis

*Salmonella* prevalence with their 95% confidence level were determined by descriptive statistical analysis. The comparison of *Salmonella* positive proportion among their relevant specifications (location or meat type) were considered using fisher's exact test. Epi Info™ version 7 completed all analyses. Statistically significant levels were determined at $p < 0.05$.

## RESULTS

A total of 370 raw meat samples were collected on 13 sampling days over a period of 5 months. This included 135 samples from six retail markets in Nong Khai Province, Thailand and a further 235 samples from five retail markets in Vientiane Capital, Lao PDR. Samples were collected from three different types of meat, composed of samples from pork ($n = 121$), chicken ($n = 133$) and beef ($n = 116$).

The overall prevalence of *Salmonella* spp. was found to be 62.70% (232/370; 95% CI [57.67–67.48%]). Prevalence rates were significantly higher in the samples collected from Laos PDR (70.21%; 165/235) compared to those collected from the Thai sampling sites (49.63%; 67/135) ($p < 0.05$). *Salmonella* spp. prevalence was highest in pork (72.73%; 88/121) compared to chicken (55.64%; 74/133) and beef (60.34%; 70/116). The higher rate samples recovered from pork was statistically significant when compared to those recovered from chicken meats ($p < 0.05$), but there was less confidence in differences in sample recovery from beef ($p = 0.05$). Table 1 demonstrates the distribution details of

**Table 1** Distribution of prevalence and a 95% confidence interval of *Salmonella* isolated from various meat types in the Thai-Lao border area.

| Type | Location | | TOTAL |
| --- | --- | --- | --- |
| | Thailand | Laos PDR | |
| **Pork** | 18/28 | 70/93 | 88/121[A] |
| | (64.29; 44.07–81.36%) | (75.27; 65.24–83.63%) | (72.73; 63.88–80.43%) |
| **Chicken** | 31/66 | 43/67 | 74/133[B] |
| | (46.97; 34.56–59.66%) | (64.18; 51.53–75.53%) | (55.64; 46.78–64.25%) |
| **Beef** | 18/41 | 52/75 | 70/116[AB] |
| | (43.90; 28.47–60.25%) | (69.33; 57.62–79.47%) | (60.34; 50.84–69.31%) |
| **TOTAL** | 67/135[a] | 165/235[b] | 232/370 |
| | (49.63; 40.92–58.36%) | (70.21; 63.92–75.98%) | (62.70; 57.67–67.48%) |

**Notes.**

Use of fisher's exact analysis and difference of superscript (A, B) indicate significant differences ($p < 0.05$) of prevalence detected among meat types. Difference of superscript (a, b) indicates significant differences ($p < 0.05$) of prevalence detected among all locations.

*Salmonella* spp. positives with 95% confidence intervals among the different locations and sample types.

In total, 53 *Salmonella* spp. serotypes were identified by serum-agglutination according to the White-Kauffmann-Le Minor scheme (Table 2). The most common serotypes were *S.* London (26.99%; 61/232), followed by *S.* Corvallis (13.79%; 32/232), *S.* Rissen (6.47%; 15/232) and *S.* Weltevreden (6.03%; 14/232), respectively. These four serotypes were common in both countries, with the exception of *S.* London which was only identified twice in the Thai samples. Stratifying for each meat type, *S.* London was the most common serotype detected from the pork ($n = 34$) and beef samples ($n = 26$). For the chicken samples, the highest degree of frequency was found for *S.* Corvallis ($n = 27$). Interestingly, there were just four serotypes, *S.* London, *S.* Rissen, *S.* Corvallis and *S.* Typhimurium, that were distributed among all meat types.

All 232 *Salmonella* strains were submitted for antimicrobial susceptibility testing. Consequently, 76 of them (32.75%) were classified as multidrug resistant strains. Resistance to tetracycline (49.57%; 115/232) was found in the highest frequency, followed by ampicillin (35.34%; 82/232), and sulfamethoxazole-trimethoprim (28.45%; 66/232), respectively. On the other hand, 94 strains (40.52%) were found to be susceptible to all of the tested antimicrobials. Additionally, only 3 strains (1.29%) were found to be resistant to cefotaxime, while 5 (2.16%) and 6 (2.59%) strains were found to be resistant to norfloxacin and ciprofloxacin, respectively. In Laos PDR, rates of resistance were also highest against tetracycline (57.78%; 95/165), ampicillin (34.55%; 57/165) and sulfamethoxazole-trimethoprim (32.12%; 53/165). In Thailand, the resistant rates were ranked as follows; ampicillin (37.31%; 25/67), tetracycline (29.85%; 20/67) and streptomycin (20.90%; 14/67) (Fig. 3). Resistance rates differed between meat sources, especially in the most effective antimicrobials, for instance almost all strains that were resistant against norfloxacin or ciprofloxacin were from chicken meat (Fig. 3). Thus related with the individual data, there were three strain resisted at least seven antimicrobials test

**Table 2  Sero-distribution of *Salmonella* isolated from various meat types at the Thai-Lao border area.**

| *Salmonella* serotype | Nong Khai, Thailand | | | Vientiane, Laos PDR | | | Total | |
|---|---|---|---|---|---|---|---|---|
| | Pork | Chicken | Beef | Pork | Chicken | Beef | n | % |
| *S.* Agona | | | | | 2 | | 2 | 0.86 |
| *S.* Albany | | 1 | | | | | 1 | 0.43 |
| *S.* Altona | | | | 2 | | | 2 | 0.86 |
| *S.* Amsterdam | | | | 2 | | 1 | 3 | 1.29 |
| *S.* Anatum | 3 | | | 2 | | | 5 | 2.16 |
| *S.* Bareilly | | 3 | | | | | 3 | 1.29 |
| *S.* Bovismorbificans | | | 2 | | | | 2 | 0.86 |
| *S.* Brimingham | | | | 1 | | | 1 | 0.43 |
| *S.* Brunei | | 1 | | | 1 | 3 | 5 | 2.16 |
| *S.* Cerro | | | 1 | | | | 1 | 0.43 |
| *S.* Corvallis | | 8 | 1 | 4 | 19 | | 32 | 13.79 |
| *S.* Duesseldorf | | 1 | | | | | 1 | 0.43 |
| *S.* Eastbournc | | | 2 | | | | 2 | 0.86 |
| *S.* Elisabethville | | | | | | 1 | 1 | 0.43 |
| *S.* Enteritidis | | 1 | | | | | 1 | 0.43 |
| *S.* Farehan | | | 1 | | | | 1 | 0.43 |
| *S.* Farsta | | | | | 1 | | 1 | 0.43 |
| *S.* Fulda | | | 1 | | | 1 | 2 | 0.86 |
| *S.* Gabon | | | | | 1 | | 1 | 0.43 |
| *S.* Give | 1 | | | 4 | | | 5 | 2.16 |
| *S.* Goma | | | | 1 | | | 1 | 0.43 |
| *S.* Havana | | | | 2 | | | 2 | 0.86 |
| *S.* Hvittingfoss | | 2 | | 2 | 1 | | 5 | 2.16 |
| *S.* Itami | | | | | 2 | | 2 | 0.86 |
| *S.* Jerusalem | 1 | | | | | | 1 | 0.43 |
| *S.* Kapemba | | | | | 1 | | 1 | 0.43 |
| *S.* Kedougou | | | | 8 | 1 | | 9 | 3.88 |
| *S.* Kikoma | | | | | | 1 | 1 | 0.43 |
| *S.* Kortrijk | | 1 | | | | | 1 | 0.43 |
| *S.* Lexington | | | 1 | | | | 1 | 0.43 |
| *S.* Livingstone | | | | | 1 | | 1 | 0.43 |
| *S.* Lomita | | | | | 1 | | 1 | 0.43 |
| *S.* London | 2 | | | 32 | 1 | 26 | 61 | 26.99 |
| *S.* Mbandaka | | 2 | | | 1 | | 3 | 1.29 |
| *S.* Meleagridis | | | 1 | 3 | | | 4 | 1.72 |
| *S.* Mikamasima | 1 | | | | | | 1 | 0.43 |
| *S.* Monschui | | | | | 3 | | 3 | 1.29 |
| *S.* Montevideo | | | | | | 1 | 1 | 0.43 |
| *S.* Muenster | 1 | | | | | 1 | 2 | 0.86 |
| *S.* Newport | | 2 | | 1 | | | 3 | 1.29 |

**Table 2** (*continued*)

| *Salmonella* serotype | Nong Khai, Thailand | | | Vientiane, Laos PDR | | | Total | |
| --- | --- | --- | --- | --- | --- | --- | --- | --- |
| | **Pork** | **Chicken** | **Beef** | **Pork** | **Chicken** | **Beef** | **n** | **%** |
| *S.* Ordonez | | 4 | | | | | 4 | 1.72 |
| *S.* Planckendael | | | | 1 | | | 1 | 0.43 |
| *S.* Regent | | | | | 1 | | 1 | 0.43 |
| *S.* Rissen | 5 | | | 3 | 3 | 4 | 15 | 6.47 |
| *S.* Ruzizi | | | 1 | | | | 1 | 0.43 |
| *S.* Saintpaul | | 2 | | | | | 2 | 0.86 |
| *S.* Sangera | | | | | | 2 | 2 | 0.86 |
| *S.* Stanley | | 1 | | 1 | | | 2 | 0.86 |
| *S.* Stanleyville | | 1 | | | | | 1 | 0.43 |
| *S.* Typhimurium | 3 | | 1 | 2 | 3 | 2 | 11 | 4.74 |
| *S.* Uganda | | 1 | | | | | 1 | 0.43 |
| *S.* Wagenia | | | | | | 1 | 1 | 0.43 |
| *S.* Weltevreden | | | 6 | | 1 | 7 | 14 | 6.03 |
| **Total** | 18 | 31 | 18 | 70 | 43 | 52 | 232 | 100.00 |

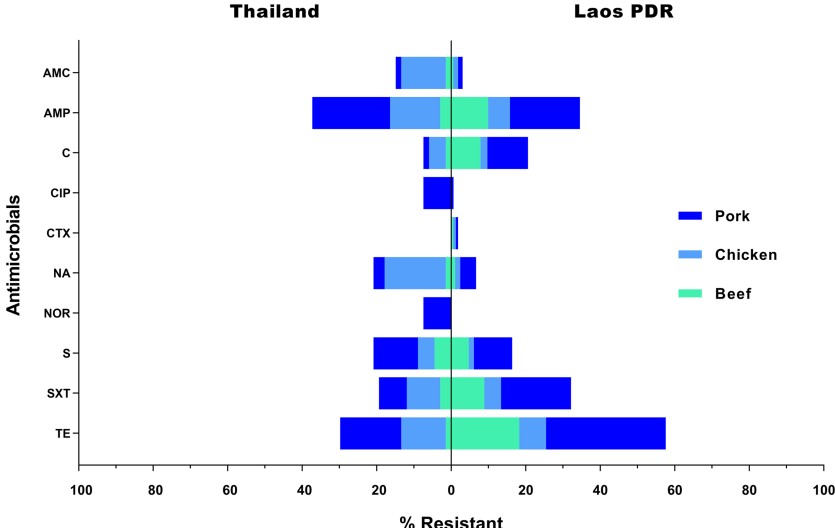

**Figure 3  Rate of resistance (%) to selected antimicrobials in *Salmonella* isolated from various meat products in the Thai-Lao border area.** Antibiotic abbreviations: amoxicillin-clavulanic acid (AMC); ampicillin (AMP); chloramphenicol (C); ciprofloxacin (CIP); cefotaxime (CTX); nalidixic acid (NA); norfloxacin (NOR); sulfamethoxazole-Trimethoprim (SXT); streptomycin (S); tetracycline (TE).

(TCH69: AMP / AMC / CIP / NA / NOR / SXT / TE, TCH44: AMP / AMC / C / CIP / NA / NOR / SXT / TE and TCH32: AMP / AMC / C / CIP / NA / NOR / S / SXT / TE) (Table S2), all three were isolated from Nong Khai, Thailand.

Figure 4 demonstrated the genetic relatedness of *Salmonella* currently being detected at the Thailand-Laos border area. From the 38 *Salmonella* strains analysed, 35 genetic characters were found. Many of the strains (*n* =28) could not be assigned to a known ST

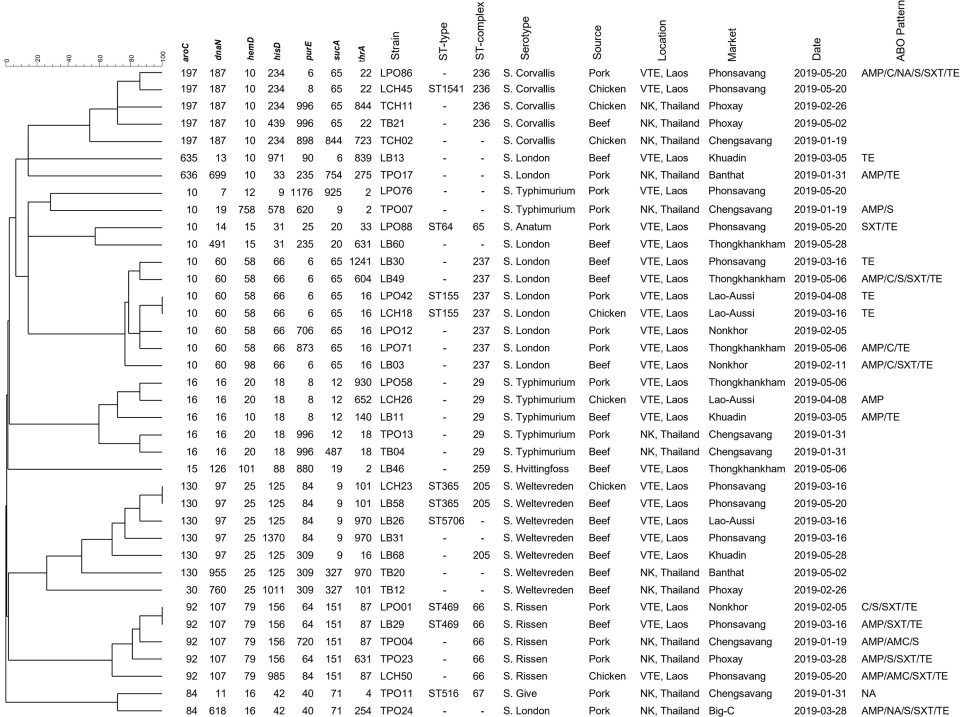

**Figure 4** Dendrogram generated using UPGMA algorithms based on MLST profiles, including phenotypic characterization and the epidemiological meta data of *Salmonella* isolated in the Thai-Lao border area.

and 11 of them could not be grouped in any previously identified ST- clonal complex. One to five variations of housekeeping genes of those 28 un-identifies from known ST were displayed in Table 3. For example, in the details, strain LPO12 with the closest of ST155 sequence type demonstrated the variation of *pur*E gene to the allelic number 706. Besides, strain LCH50 with the closest of ST469 demonstrated the variation of *his*D and *pur*E to the allelic number of 985 and 84, respectively. The However, for the 7 known STs (ST64, ST155, ST365, ST469, ST516, ST1541, ST5706), four of those (ST64, ST516, ST1541, ST5706) were distinct to a single strain. Nonetheless, two strains were grouped in the ST155, ST365 and ST 469. From the displayed in Fig. 4, strains grouped in ST155 were derived from a Lao-Aussi market, one (LCH18; 16 Mar) was chicken meat and the another (LPO42; 8 Apr) was pork. Two strains of ST365 originated from Phonsavang market, one (LCH23; 16 Mar) was chicken and the another (LB58; 20 May) was beef were also identified. Additionally, the strains grouped in ST469 were originated from pork (LPO01; 2 May) and beef (LB29; 16 Mar) samples collected from different markets located in nearby areas.

## DISCUSSION

The findings obtained from this study represent scientific information on the burden and intensity of *Salmonella* in livestock meats in a 30 km radius surrounding the Thai-Lao

**Table 3  Variation of housekeeping genes loci of un-identified Sequence Type (ST) Salmonella strains circulating in the Thai-Lao border area.**

| Strain | Most related ST[a] | 7 Housekeeping gene for MLST[b] | | | | | | |
|---|---|---|---|---|---|---|---|---|
| | | *aro*C | *dna*N | *hem*D | *his*D | *pur*E | *suc*A | *thr*A |
| LPO12 | ST155 | 10 | 60 | 58 | 66 | 6 → 706 | 65 | 16 |
| LPO58 | ST29 | 16 | 16 | 20 | 18 | 8 | 12 | 18 → 930 |
| LPO71 | ST155 | 10 | 60 | 58 | 66 | 6 → 873 | 65 | 16 |
| LPO76 | ST7498 | 10 | 7 | 12 | 9 | 1176 | 9 → 925 | 2 |
| LPO86 | ST1541 | 197 | 187 | 10 | 234 | 8 → 6 | 65 | 22 |
| LCH26 | ST29 | 16 | 16 | 20 | 18 | 8 | 12 | 18 → 652 |
| LCH50 | ST469 | 92 | 107 | 79 | 156 → 985 | 64 → 84 | 151 | 87 |
| LB03 | ST155 | 10 | 60 | 58 → 98 | 66 | 6 | 65 | 16 |
| LB11 | ST29 | 16 | 16 | 20 → 10 | 18 | 8 | 12 | 18 → 140 |
| LB13 | ST1799 | 202 → 635 | 4 → 13 | 10 | 33 → 971 | 90 | 6 | 275 → 839 |
| LB30 | ST155 | 10 | 60 | 58 | 66 | 6 | 65 | 16 → 1241 |
| LB31 | ST5706 | 130 | 97 | 25 | 125 → 1370 | 84 | 9 | 970 |
| LB46 | ST446 | 15 | 126 | 101 | 88 | 8 → 880 | 19 | 18–2 |
| LB49 | ST155 | 10 | 60 | 58 | 66 | 6 | 65 | 16 → 604 |
| LB60 | ST64 | 10 | 14 → 491 | 15 | 31 | 25 → 235 | 20 | 33 → 631 |
| LB68 | ST365 | 130 | 97 | 25 | 125 | 84 → 309 | 9 | 16 |
| TPO04 | ST469 | 92 | 107 | 79 | 156 | 64 → 720 | 151 | 87 |
| TPO07 | ST34 | 10 | 19 | 12 → 758 | 9 → 578 | 5 → 620 | 9 | 2 |
| TPO13 | ST29 | 16 | 16 | 20 | 18 | 8 → 996 | 12 | 18 |
| TPO17 | ST3157 | 636 | 4 → 699 | 10 | 33 | 90 → 235 | 6 → 754 | 275 |
| TPO23 | ST469 | 92 | 107 | 79 | 156 | 64 | 151 | 87 → 631 |
| TPO24 | ST616 | 84 | 11 → 618 | 16 | 42 | 40 | 71 | 4 → 254 |
| TCH02 | ST197 | 197 | 187 | 10 | 234 | 8 → 898 | 844 | 22 → 723 |
| TCH11 | ST1541 | 197 | 187 | 10 | 234 | 8 → 996 | 65 | 22 → 844 |
| TB04 | ST29 | 16 | 16 | 20 | 18 | 8 → 996 | 12 → 487 | 18 |
| TB12 | ST283 | 101 → 30 | 97 → 760 | 25 | 86 → 1011 | 101 → 309 | 19 → 327 | 101 |
| TB20 | ST5706 | 130 | 97 → 955 | 25 | 125 | 84–309 | 9 → 327 | 970 |
| TB21 | ST1541 | 197 | 187 | 10 | 234 → 439 | 8 → 996 | 65 | 22 |

[a]The most genetic relateness Sequence Type (ST) with the untypable strain.
[b]Variation of the housekeeping genes alleic number from known ST to untyable ST.

Friendship Bridge I (Fig. 2). It is well known that this bridge is the primary place for the expansion of trade that has occurred along the Thailand and Laos PDR border since 1994. Overall, prevalence of *Salmonella* spp. in all sampled meat were found to be 63%. Pork had the highest prevalence (73%), followed by beef (60%) and just over half of the chicken meats (56%) tested were contaminated with *Salmonella* (Table 1). Contamination of pork was higher than previously observed in a similar study by *Sinwat et al. (2016)*, where 65% (95% CI [59.26–70.47]%) of pork were contaminated. However, for beef samples the degree of prevalence in this study was lower than the results of a study conducted by *Boonmar et al. (2013)*, in which the prevalence was recorded at 82% (95% CI [58.97–93.81]). No studies have reported on the prevalence of *Salmonella* contamination in chicken meats sold in this region. A study conducted by *Trongjit et al. (2017)* reported on the degree of prevalence of

*Salmonella* infection at the Thai-Cambodia border, which was found to be 73.43% (95% CI [66.65–79.42]), which is higher than our observed results. In general, most studies have not demonstrated much difference when reporting numbers, which has been proven by an overlap in the 95% confidence intervals. Variations in the results might depend upon the time period of the sampling along with any existing geographical factors. Nevertheless, high levels of *Salmonella* contamination still persist, even though some intervention has been implemented in some of these areas. *Salmonella* remains a problem in the meat being sold in this region, which has been an important public health concern for the last half decade.

The samples isolated from meat sold in Laos PDR were statistically and significantly higher than those isolated over the border in Thailand (Table 1). Normally, sanitation practices are different at each location. In fact, there is no supermarket in Laos DPR. All fresh food, such as meat, can only be bought from fresh markets and mini-grocery stores. The lack of covering materials, unsuitable storage conditions and inadequate disinfection practices at the purchasing areas during the meat cutting and handling processes can substantially increase the risk of bacterial colonization (*Gomes-Neves et al., 2012*; *Patchanee et al., 2016*). In Thailand, supermarkets are the preferred place to purchase meat. One-sixth of the Thai samples collected in our study were obtained from supermarkets. Supermarkets provide stringent regulation of their facilities and implement biosecurity and hygiene policies along with quality control practices at the pre-harvesting and harvesting stages along the supply chain. However, in general, standard protocols for every type of retail outlet (supermarkets, mini-grocery stores and fresh markets) tend to be higher in larger cities (*Trongjit et al., 2017*).

With a focus on meat type, pork was found to be the most prevalent when compared with the other types of meat. The reason for this may not be clear. It could be due to the high bacterial loads that emerge during previous production stages, which can then lead to instances of contamination at retail outlets. According to the information obtained from previous studies, *Salmonella* prevalence at pig farms and during pig slaughtering processes was higher than in chicken and beef production (*Padungtod & Kaneene, 2006*; *Trongjit et al., 2017*; *Phongaran, Khang-Air & Angkititrakul, 2019*). Another possible explanation for this is that pork is the most common type of meat consumed in this region (*Napasirth & Napasirth, 2018*). The amount of dressing required, along with any other forms of manipulation before meat is sold, would likely increase the opportunity for contamination or re-contamination by product exposure at the final.

Based on the findings of previous studies, *S.* Typhimurium and *S.* Rissen are the majority serotypes identified in pork (*Patchanee et al., 2016*; *Sinwat et al., 2016*). For chicken meat, *S.* Corvallis and *S.* Enteritidis are known to be the dominant serotypes (*Trongjit et al., 2017*). Additionally, *S.* Stanley and *S.* Typhimurium have been reported as the most commonly recorded serotypes in beef (*Boonmar et al., 2013*). At the moment, these serotypes have not been universally matched to each meat type, but low frequencies were recorded in some instances. However, *S.* Corvallis is still noted as being the dominant serotype in chicken meat (Table 2). Consequently, new typical sero-characteristics for this region should be set for pork and beef. As the data indicates, *S.* London is presently the most common

serotype. Furthermore, several serotypes have been detected, such as *S.* Altona, *S.* Cerro, *S.* Elisabethville, *S.* Itami, *S.* Mikamasima, *S.* Ruzizi, etc. They have been reported for the very first time about the isolation of *Salmonella* in the region. Time factor and sample picking and the cross-contamination that occurs from other sources, are notable factors. Furthermore, even though the same serotype might be presented at several meat origins, it cannot be concluded that two or three meat types would represent a sharing pool for *Salmonella* identified from similar sources (*Sinwat et al., 2016*). All of which would need to be proven.

*Salmonella* specimens isolated in this study display a relatively high frequency of resistance against tetracycline and ampicillin (Fig. 3), which is consistent with the findings of previous investigations in the region (*Padungtod & Kaneene, 2006*; *Pulsrikarn et al., 2012*; *Boonmar et al., 2013*). From the past until now, the antimicrobials have been widely used for treatment and prophylaxis in livestock. However, excessive or inappropriate use is considered to be a critical factor that has led to the current situation of resistance (*Jajere, 2019*). On the contrary, low resistance rates have been recorded for cefotaxime, norfloxacin and ciprofloxacin. Instances of resistance to these antimicrobials are particularly important as these are the drugs of choice for treatment of human salmonellosis. Specifically, quinolones (norfloxacin and ciprofloxacin) are now often used as the first line of treatment (*Kurtz, Goggins & McLachlan, 2017*). All quinolone-resistant strains isolated in these study areas originated from chicken meat collected in Thailand, and all were found to be resistant to 7-9 of the tested antimicrobials. Selective pressures such as those associated with antimicrobial use, unsuitable temperatures or pH levels, could be considered sub-lethal stress factors during the short production cycle in broiler farms. Additionally, the characterizing of bacterial community composition in chicken's gut should be taken to determine the mechanisms of action on the resistome (*Shang, Wei & Kang, 2018*; *Yang et al., 2019*).

The strains isolated in this study were genetically diverse and five-sixths could not be assigned to a previously described ST. Regional diversity following microevolution or mutation is a possible explanation for the finding (*Harbottle et al., 2006*; *Liu et al., 2011*). Variation in 3-5 of the housekeeping genes used to determine ST was also observed (Table 3), meaning that those strains could not be assigned to any clonal complex either. Large scale, shared genetic relatedness of *Salmonella* strains isolated along the Thai-Lao border was not observed (Fig. 4). The finding infers that the transboundary food supply chain from locations ahead of the marketplace (farm and/or slaughterhouse) is not involved in contamination. A larger scale analysis, including all strains previously collected in the region and deposited in MLST databases would help identify additional shared reservoirs of contamination that may have been missed in our study.

*Salmonella* genotypes isolated from a single market also demonstrated high levels of diversity, nine and six *Salmonella* genotypes were isolated from various meat types in markets from Phonsavang and Chengsavang, respectively. This suggests that more than one infection source can exist at a given location. Contamination can occur by itself in the purchasing area or as a result of inadequate processing when the meat product may

have already been contaminated. Unhygienic practices at previous production sites, such as on farms and at slaughterhouses, or at transportation hubs may increase transmission (*Heyndrickx et al., 2002*; *Campioni, Bergamini & Falcao, 2012*). Clones of sequence types ST155 and ST365 were detected on different sampling dates from the same marketplaces, indicative of persistent or residential *Salmonella* contamination. Additionally, two strains of ST469 were recovered from different markets in nearby areas, hinting at potential shared supply routes and their role in dissemination of *Salmonella*.

## CONCLUSIONS

This is the first comprehensive study for the transmission dynamics of *Salmonella* in Thailand-Lao DPR border area. Multi Locus Sequence Typing (MLST) analysis could eventually serve to pinpoint the genotypic characterization differences and provide more crucial information than other previous studies such prevalence study and phenotypic description. It has been successful for the characterization of clonal relationship among *Salmonella* circulating in meat sold in the middle Mekong basin area. To the best of our knowledge, there is no molecular evidence of transboundary epidemiological link among these MLST sequence types. The findings obtained from the study indicate that extensive quality control checking at pre-retail stages should be strictly implemented. Likewise, regular disinfecting of all equipment, as well as at working areas, must be applied. *Salmonella*'s presence was relatively high in terms of prevalence and highlight multidrug resistant for one-third of them. Standard hygienic protocols and integrating of knowledge/attitude/practices in antimicrobial using are known to be maintained at a higher level in administrative areas. However, to expand the epidemiological knowledge of the pathogen, analysis of the *Salmonella* genetics together with the geographic matching-strains submitted in MLST database should be completed for the further whole genome sequencing study. Finally, future efforts in strengthening food safety education and awareness programs would help authorities to establish strategies that could potentially reduce the transmission of *Salmonella* and other foodborne pathogens to downstream consumers.

## ACKNOWLEDGEMENTS

The authors would like to thank the Bacteriology Section, Veterinary Research and Development Center (Upper Northern Region), Lampang, Thailand and the Vientiane Capital Agriculture and Forestry Department, Vientiane, Laos PDR for their valuable contributions. We would like to thank the Department of Veterinary Medicine, Faculty of Agriculture, National University of Laos for allowing us to use their laboratory facilities for the purposes of diagnosis. Finally, we would like to thank the people who operate the http://enterobase.warwick.ac.uk/species/senterica/allele_st_search website which made our regional analysis possible.

### Funding

This work was supported by the Faculty of Veterinary Medicine, Chiang Mai University (Project ID: CMU-MIS R000017502). The funders had no role in study design, data collection and analysis, decision to publish, or preparation of the manuscript.

### Grant Disclosures

The following grant information was disclosed by the authors:
Faculty of Veterinary Medicine.

### Competing Interests

The authors declare there are no competing interests.

### Author Contributions

- Dethaloun Meunsene conceived and designed the experiments, performed the experiments, analyzed the data, prepared figures and/or tables, and approved the final draft.
- Thanaporn Eiamsam-ang analyzed the data, prepared figures and/or tables, and approved the final draft.
- Prapas Patchanee and Phacharaporn Tadee conceived and designed the experiments, authored or reviewed drafts of the paper, and approved the final draft.
- Ben Pascoe analyzed the data, authored or reviewed drafts of the paper, and approved the final draft.
- Pakpoom Tadee conceived and designed the experiments, performed the experiments, analyzed the data, authored or reviewed drafts of the paper, and approved the final draft.

### DNA Deposition

The following information was supplied regarding the deposition of DNA sequences:
The Salmonella housekeeping gene sequences are available at GenBank: MW198823–MW199053.

### Data Availability

The raw data are available in the Supplemental File.

### Supplemental Information

Supplemental information for this article can be found online at http://dx.doi.org/10.7717/peerj.11255#supplemental-information.

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
