# Peer review of "Molecular evidence for cross boundary spread of Salmonella spp. in meat sold at retail markets in the middle Mekong basin area"

_PeerJ, doi:10.7717/peerj.11255_

## Round 0.1 · original submission · Major Revisions

Authors, please find enclosed the comments from reviewers. You would see that decision from reviewers was wide-ranged, from acceptance to major revisions.

You are encouraged to consider their comments carefully, address them diligently because it will surely improve the quality of this work.
Authors, kindly consider the following as well:

a) The study is not well justified. In the final paragraph of introduction, prior to starting the objective, kindly present the rationale for this study. Why is it needful to determine the prevalence of Salmonella in meat sold at retail markets in the middle Mekong Basin area? What is so unique about the border of Thailand and Lao PDR, surrounding the Thai-Lao Friendship Bridge? Even though you have stated that the chances of pathogen transmissions between borderline communities are likely to be high, that is not sufficient. Kindly think about these.

b) Methods...why did you use the Win Epi online program, to determine the sample size of the study. Kindly indicate other published studies that have used it elsewhere. Why it is essential to use this software/platform?

Why were rates of 65% (Sinwat et al., 2016), 73% (Trongjit et al., 2017) and 82% (Boonmar et al., 2013) respectively chosen to calculate the sample sizes for pork, chicken and beef? Why were 8.5% and 95% selected as the “accepted error” and “confidence level” values, respectively for the required feature inputs?

c) It is quite clear from results and discussion that statistics was performed.

In the methods, the last part should be the statistical create a subsection statistical analysis., and carefully articulate all the statistical aspects.

d) In the discussion, please kindly insert (Refer to Table X/Figure X) to draw the attention of reader to that specific figure/table.

This statement 'The prevalence of Salmonella in the Laos PDR-originated samples was statistically and significantly high (Lines 262-263)', what do you mean? Where is the statistics indicating it?

Authors have performed a very brilliant study . I look forward to your revised manuscript. Thank you for finding PeerJ as your suitable journal of choice.

·

Basic reporting

The authors need to improve on their sentencing so as not to make them ambiguous. Their writing style should be such that will enable readers understand what they mean and the information they are conveying. I have tried to indicate the areas in the manuscript.

Experimental design

The design is okay, however they need to give details of some of the methods they used in the study. I must commend them, its a good work. Their study was in-depth enough.

Validity of the findings

The findings are meaningful, however Table 1 may require a little more statistical.
The data will be more explanatory if they could assign superscripts to each of the values. This will help one to see the values that are significantly different at a glance. Also indicate whether the statistical comparison is along column or row.

Reviewer 2 ·

Basic reporting

no comment

Experimental design

no comment

Validity of the findings

no comment

Additional comments

In this work, the authors focused their attention on the Salmonella contamination in livestock products that are very common in some areas of the world and specifically in the characteristics markets of the West Asia.
The manuscript is well written.
The paper is supported by relevant experimental data, the presentation is clear and most of the conclusions are justified.
I recommend it for publication in this journal.

·

Basic reporting

The authors have conducted an important study on Salmonella spp. which causes salmonellosis in human.
The manuscript is clear but requires attention for English used. I have worked through the introduction but requires improvement in other parts.
Literature references, sufficient field background/context have been provided.
Structure of the article is correct. Improvements in tables and figures are required. Table 3 requires attention and figure 2 could be more illustrative (than descriptive) and colourful for establishing useful knowledge from it.

Besides language, the major concerns are lack of illustrations (figures) in the manuscript.
Additionally, since the sequencing data of the strains is available, an analysis of sequence comparison is highly recommended to increase the impact of the study.
Circos plots for comparison of various genes, gene annotations, intron-exon annotations of genes, and protein models and homology modelling are few examples of the study that could be included bases on sequence data available with the authors.

Detailed comments are attached as pdf.

Experimental design

No comment.

Validity of the findings

No comment.

Additional comments

Refer to comments in part 1.

Reviewer 4 ·

Basic reporting

Manuscript submitted by Menusene et al., intend to investigate the prevalence of Salmonella species in meat sold at retail markets in the middle Mekong basin area. Authors have utilized Multi-locus sequence typing to genotype several Salmonella serotypes isolated from 370 Meat samples during January to May 2019 from five retail markets in Vientiane Capital, Lao PDR and six retail markets in Nong Khai Province, Thailand.

Comments:
1) Title of manuscript is ambiguous and confusing. Authors should consider rephrasing the title based on findings in the manuscript.

2) Authors have mentioned that " Rates of 65%, 73% and 82% were chosen to calculate the sample sizes for pork, chicken and beef, respectively". Authors should elaborate how they defined "expected prevalence" and calculated sample size for this study.

3) Manuscript suffers a lot from grammatical errors and spelling mistakes. Authors should get the manuscript proofread by a native speaker to ensure that an international audience can clearly understand the text and findings.

4) Throughout the manuscript, authors missed to italicize the species name. For example, Line 39/ 40: S.London, S. Corvallis, S.Rissen. Scientific name of the species is always italicized without exception.

Experimental design

Question addressed by authors in the study is well defined and relevant. Antimicrobial resistance is a pertinent problem and addressing Salmonella resistance in livestock meats around Thai-Lao friendship bridge highlights importance of stringent food safety regulation and implementation of standard hygiene protocol in pre-retail meat industry.

Comments:

1) Authors mentioned in line 147 that " an association between a Salmonella positive proportion and the relevant specifications were determined by descriptive statistical analysis" what kind of statistical analysis they are referring here ?

2) What were the Primers, PCR conditions used for multilocus sequence typing? Authors should provide sufficient details in methods section so that findings can be replicated and reproduced.

Validity of the findings

1) Underlying data have been provided with proper control.

2) Conclusions are well stated and linked to original research question addressed.

Additional comments

Manuscript submitted by Menusene et al., intend to investigate the prevalence of Salmonella species in meat sold at retail markets in the middle Mekong basin area. Authors have utilized Multi-locus sequence typing to genotype several Salmonella serotypes isolated from 370 Meat samples during January to May 2019 from five retail markets in Vientiane Capital, Lao PDR and six retail markets in Nong Khai Province, Thailand. Question addressed by authors in the study is well defined and relevant. Antimicrobial resistance is a pertinent problem and addressing Salmonella resistance in livestock meats around Thai-Lao friendship bridge highlights importance of stringent food safety regulation and implementation of standard hygiene protocol in pre-retail meat industry.


Comments:
1) Title of manuscript is ambiguous and confusing. Authors should consider rephrasing the title based on findings in the manuscript.

2) Manuscript suffers a lot from grammatical errors and spelling mistakes. Authors should get the manuscript proofread by a native speaker to ensure that an international audience can clearly understand the text and findings.

3) Authors have mentioned that " Rates of 65%, 73% and 82% were chosen to calculate the sample sizes for pork, chicken and beef, respectively". Authors should elaborate how they defined "expected prevalence" and calculated sample size for this study.

4) Throughout the manuscript, authors missed to italicize the species name. For example, Line 39/ 40: S.London, S. Corvallis, S.Rissen. Scientific name of the species is always italicized without exception.

5) Authors mentioned in line 147 that " an association between a Salmonella positive proportion and the relevant specifications were determined by descriptive statistical analysis" what kind of statistical analysis they are referring here ?

6) What were the Primers, PCR conditions used for multilocus sequence typing? Authors should provide sufficient details in methods section so that findings can be replicated and reproduced.

---

## Round 0.2 · Minor Revisions

Reviewers have attended to your revised manuscript. You can see two indicated accept, and one indicated minor revisions. In addition, please, Editor requires your important attention to address the following:

a) Introduction section:
-After the first paragraph, given the context of this work, the next (new) paragraph should deal with cross boundary spread of microorganisms in markets in general. What is the underlying concept behind cross boundary spread of microorganisms? What are the causes, or factors that lead to it? Why does it occur? What are the public health implications? Editor envisages 6-7 sentences here to address this.
-In the last paragraph of introduction, please, before stating the purpose of this study, kindly use one or two sentences to reiterate the knowledge gap/research question that underpins the premise of this study.

b) Materials and methods
-Please, kindly start this section new subsection captioned: 'overview of experimental program'
This must provide a snapshot of the entire study, using a schematic flow diagram that includes: Determination of sample size> Identification of retail markets> Collection of pork, chicken and beef samples > Labelling, packaging and refrigerated storage> Laboratory analysis {divided into Salmonella isolation and identification/Serotyping and Antimicrobial susceptibility testing/Multilocus sequence typing (MLST)...apply your discretion to schematically represent this , showing diagrammatically how samples were allocated, etc. Describe this schematic flow very tersely. This will add value to this work, and help readers follow it
- The rest of the materials and methods is very good
c) Results is very good
d) Discussion is also ok, however, please, kindly insert (Refer to Fig ??, or Table ??) in all the areas where specific Figures or Table information , already in the results section, but now being discussed, is mentioned. This has to be there, to help ensure readers follow the discussion ok. Editor will be look out for this , in your revised manuscript.

e) Conclusions: It is quite ok, however not so strong. Ask yourselves this question:
-Was the objective(s) achieved, if yes, where , and how, and which is one stands out, that makes this work important. Reiterate what the specific objective of study was, and why this study was conducted.
-Please, brainstorm on what you might perceive as a potential limitation (if you deem there is none, exclude this)
-Please reinforce more clearly the direction of future studies, that will help supplement existing information.

This is very scholarly study. Look forward to your revised manuscript.

·

Basic reporting

They have improved on their writeup compared to their previous submission. However, there are a few technical terms they need to correct. Some have been indicated eg they should replace ‘collect’ or ‘collected’ with ‘isolate’or ‘isolated’. They should also write the names of the microorganisms as approved internationally, the first name begins with upper case letter while the second name begins with lower case letter. They need to correct these in the entire manuscript. If these are corrected the paper can be accepted.

Experimental design

The design is good and the supplied enough information in the methodology. The study was in depth enough and rigorous, and the methods used were appropriate.

Validity of the findings

Their work has provided important information as regards sources of contamination and spread of Salmonella through meat sold in the region. They were able for the first time to isolate and identify some sertypes which before now had not been isolated.

Additional comments

I have said all I want to say.

·

Basic reporting

The authors have improved the manuscript well and appear to be fit for publication.

Experimental design

no comment

Validity of the findings

no comment

Additional comments

Despite the suggestion of making the figure 2 more illustrative, the authors want to stick to the original figure. I would suggest the authors for future manuscripts use more illustrative figures as they attract the audience increasing reads and thus citations.
Finally, since the authors plan not to use further analysis of the sequencing data in the present manuscript, plan a separate manuscript using this important data resource that has been generated.

Reviewer 4 ·

Basic reporting

Concerns has been satisfactorily addressed.

Experimental design

Concerns has been satisfactorily addressed.

Validity of the findings

Concerns has been satisfactorily addressed.

Additional comments

Concerns has been satisfactorily addressed.

---

## Round 0.3 · Minor Revisions

Thank you authors for addressing the comments raised by the reviewers. You have made great efforts. However, editor suggestions appear not completely addressed.

In the materials and methods, authors did not address the overview of experimental program. it cannot start with 'The sample size of this study was determined....' You simply repeated previous information, and placed it under the caption, overview of experimental program. This is not what was suggested.

Editor kindly request to provide this overview of experimental program, as advised. This must provide a snapshot of the entire study, using a schematic flow diagram that includes: Determination of sample size> Identification of retail markets> Collection of pork, chicken and beef samples > Labelling, packaging and refrigerated storage> Laboratory analysis {divided into Salmonella isolation and identification/Serotyping and Antimicrobial susceptibility testing/Multilocus sequence typing (MLST)...apply your discretion to schematically represent this , showing diagrammatically how samples were allocated, etc. Describe this schematic flow very tersely. This will add value to this work, and help readers follow it

Line 120 should start new caption 'Sample collection procedures'

In the conclusions, the authors appear to have ignored editors suggestions. Was the objective(s) achieved, if yes, where , and how, and which is one stands out, that makes this work important. Reiterate what the specific objective of study was, and why this study was conducted.
It is not just by stating that this is the first comprehensive study of Salmonella circulating in meat sold in the middle Mekong basin area. You must lay the foundation of the the hypothesis that led to this study, what this study has been able to address, why it is different from existing literature, which is near to it, and why this one stands out, before you can make this claim. And in making this claim, you have to state 'to the best of our knowledge'


Please, kindly attend to these carefully. The editor will be looking to see these fully addressed. Thank you.

---

## Round 0.4 · Minor Revisions

Thank you for revising your manuscript. Thank you for providing a useful schematic diagram showing the overview of experimental program in the beginning of materials and methods. Kindly provide 2-3 sentence description of this schematic diagram, to demonstrate how it directly reflects the objective of this work.

In your conclusion, you state this is the first study of its kind, why and how is it the first study? Justify it not only using your findings but establishing why and how it differs from the existing published literature, and how your findings have supplemented existing information

You have done great work. Kindly address these because it helps strengthen the quality of this very good paper.
Look forward to your revised manuscript :)

---

## Round 0.5 · accepted · Accept

Thank you for revising your manuscript. It is now acceptable and can now be published. Thank you for your very fine contribution, and for finding PeerJ as your journal of choice. Looking forward to your future scholarly works. Congratulations :)